# Knowledge, attitude and practice regarding diabetes and hypertension among school students of Nepal: A rural vs. urban study

**Deekshanta Sitaula**[1☯]*, **Niki Shrestha**[2☯], **Santosh Timalsina**[3☯], **Bandana Pokharel**[4‡], **Sachin Sapkota**[4‡], **Suchita Acharya**[4‡], **Rohit Thapa**[4‡], **Aarati Dhakal**[5‡], **Sarita Dhakal**[6‡]

**1** Department of General Medicine, Rasuwa District Hospital, Dhunche, Bagmati Province, Nepal, **2** Department of Community Medicine, Chitwan Medical College, Chitwan, Bagmati Province, Nepal, **3** Research Unit, Chitwan Medical College, Chitwan, Bagmati Province, Nepal, **4** School of Medicine, Chitwan Medical College, Chitwan, Bagmati Province, Nepal, **5** Department of Community Program, Kathmandu University Hospital, Dhulikhel, Bagmati Province, Nepal, **6** Department of Emergency Medicine, Ramechhap District Hospital, Ramechhap, Bagmati Province, Nepal

☯ These authors contributed equally to this work.
‡ BP, SS, SA, RT, AD, SD also contributed equally to this work.
* drdeekshanta@gmail.com

## Abstract

### Background

The burden of non-communicable diseases like diabetes and hypertension is increasing worldwide including low-and middle-income countries. Good knowledge of such diseases among young people will make them adopt a healthy lifestyle from an early age, which will, in turn, prevent them from developing such non-communicable diseases. This study aimed to assess the knowledge, attitude, and practice of rural and urban school students regarding diabetes and hypertension. We also aimed to see the differences in the knowledge, attitude, and practice of students from rural vs. urban communities.

### Methods

A school-based cross-sectional study was conducted from May 1 2021 to June 30, 2021, in four schools in Nepal (1 from a metropolitan city, 2 from an urban municipality, and 1 from a rural municipality). The study was conducted among the secondary-level students of classes 9 and 10 in each school. The data were collected from the participants via pre-tested questionnaires and analyzed in the Statistical Packages for Social Sciences (SPSS) version 20.0. Logistic regression analysis was carried out to determine the determinants of knowledge and attitude regarding diabetes and hypertension.

### Results

Of 380 respondents, 35.5% were residents of metropolitan city, 37.4% were from the urban municipality and 27.1% were from the rural municipality. The mean age of respondents was 15.61±0.99 years and 51.1% were male. Respondents having a family history of diabetes and hypertension were 21.1% and 37.9% respectively. Respondents from the metropolitan

**Data Availability Statement:** All relevant data are within the manuscript and its Supporting Information files.

**Funding:** The author(s) received no specific funding for this work.

**Competing interests:** The authors have declared that no competing interests exist.

city had significantly higher mean knowledge scores than the respondents from the urban and rural municipality (p<0.001) while there was no significant difference in mean attitude scores. There was significantly higher daily consumption of fruits and vegetables among the participants from rural municipality (p<0.01) while no significant difference was seen in salt consumption and time spent on physical activity. In univariate regression analysis, place of residence, family occupation, parental education, and family history of diabetes and hypertension were significantly associated with good knowledge level. In multivariate analysis, only a higher grade of study (grade 10 in comparison to grade 9) was an independent predictor of a student's good attitude level.

## Conclusion

In general, there was a good attitude towards diabetes and hypertension despite poor knowledge. The mean knowledge scores were lower in urban municipality and rural municipality compared to metropolitan city. Low knowledge scores on diabetes and hypertension among the students show an urgent need for school-based interventional programs focusing on non-communicable diseases and lifestyle modification with more emphasis on rural communities.

## Introduction

Non-communicable diseases (NCDs) have emerged as a serious public health burden and the leading cause of death globally. As per WHO NCD country profiles, NCDs accounted for 71% of the 57 million deaths that occurred globally in 2016. Among the NCD deaths, 78% of total NCD deaths and 85% of premature NCD deaths occurred in low-and middle-income countries [1]. According to Global Burden of Disease (GBD) 2019, NCDs accounted for 1.62 billion healthy-years lost in 2019, with the rise of Disability Adjusted Life Years (DALYs) from 43.2 percent of total DALYs in 1990 to 63.8 percent in 2019 [2]. The major NCDs responsible for death were cardiovascular diseases (17.9 million deaths, 44% of total NCD deaths), cancers (8 million deaths, 22% of total NCD deaths), chronic respiratory diseases (3.8 million deaths, 9% of total NCD deaths) and diabetes (1.6 million deaths, 4% of total NCD deaths) [1]. Just like other low-and middle-income countries, Nepal is also facing the surging burden of NCDs, where an estimated 66% of all the deaths were due to NCDs in 2016 [1, 3]. A recent population-based study on selected NCDs in Nepal reported the prevalence of Chronic Obstructive Pulmonary Disease (COPD) to be 11.7%, Diabetes Mellitus 8.5%, chronic kidney disease 6.0%, and coronary artery disease 2.9% [4]. In a previous study from Nepal, the hospital-based NCD prevalence was reported to be 31%, with COPD being the most common NCD (43%) followed by cardiovascular diseases (40%), Diabetes Mellitus (12%) and cancer (5%). The majority of cardiovascular cases were hypertension (47%) in that study [3]. Similarly, the prevalence of hypertension among the urban adult population was 22% [5]. In 2003, the prevalence of diabetes and impaired fasting glucose in the urban Nepalese population were 14.2% and 9.1% respectively [6]. A more recent data from STEPS survey Nepal 2019 reported the prevalence of hypertension and diabetes to be 24.5% and 5.8% respectively [7].

The leading behavioral risk factors associated with four major NCDs are tobacco use, harmful use of alcohol, physical inactivity and unhealthy diet [1]. It has been reported that the poor score in behavior factors increases the risk of mortality by four-folds [8]. With increasing

urbanization in low-and middle-income countries, there is a change in lifestyle of people, shifting towards a sedentary lifestyle and unhealthy diet. This change in lifestyle leads to rise in NCDs [9]. WHO has recommended public education on the behavioral change regarding physical activity levels and dietary modification under "best buys" to reduce the growing burden of NCDs [1]. Behaviors and habits develop and become established during earlier stages of life. It is easier to change the habits and lifestyle during the early stage as it becomes more difficult after adulthood because of formed habits and automatic behaviors [10, 11]. So, promoting healthy behaviors through school settings in the early stage of life can significantly contribute to the prevention of NCDs.

In a study conducted among high school students in Thailand, 93% were familiar with Diabetes, and two-thirds of them knew it as a non-communicable disease. In the same study, 92.84% of students were familiar with hypertension and 77.32% knew it is non-communicable. 77.13% of respondents did not know that a family history of diabetes is a risk factor for diabetes while two-thirds did not notice a family history of hypertension as a risk factor [12]. Another study from Oman reported that 63% of high school students falsely believed diabetes as a curable disease [13]. In a study conducted among the general public in Southern Sri Lanka, 77% of participants had moderate or above moderate knowledge of diabetes but the attitude towards diabetes was poor in the majority (90%) [14]. A rural-urban study conducted in Malaysia reported that the rural respondents had lower scores regarding knowledge of NCDs and diabetes mellitus but they had significantly higher attitude levels [15]. In a pre-interventional study in North India regarding obesity and Diabetes, low baseline knowledge and behavior scores were reported in both government and private school children. However, after the intervention, scores improved in all the children irrespective of the types of schools [16]. It has been reported that 43.8% of diabetic patients in Nepal had insufficient knowledge of Diabetes [17]. However, to the best of our knowledge, there has hardly been any study in Nepal assessing the KAP of school students on NCDs.

This study aims to assess the knowledge, attitude, and practice regarding diabetes and hypertension among the secondary level school students in selected urban and rural areas in Nepal and analyze the difference between them.

## Methods

### Study design

Institution-based cross-sectional analytical study was designed to explore the knowledge, attitude, and practice of school students from selected rural and urban communities of Nepal regarding diabetes and hypertension. We also aimed to compare the differences in KAP among students from different local government units (i.e., metropolitan city, municipality, and rural municipality). Data were collected from May 1 2021 to June 30, 2021.

### Study settings and participants

This study was conducted in secondary level schools of different local government units in Nepal: Valmiki Shiksha Sadan of Bharatpur metropolitan city, Souvenir Boarding School, and Bhairabi Higher Secondary School of Bidur municipality, and Narayandevi Higher Secondary School of Likhu rural municipality. The study area comprised a metropolitan city, an urban municipality, and a rural municipality from two districts of central Nepal. They represent different levels of urbanization ranging from the metropolitan city (most urbanized) to the rural municipality (least urbanized). Bharatpur metropolitan city is located in Chitwan district while Bidur municipality and Likhu rural municipality are located in Nuwakot district (Table 1). In Nepal, there are 753 local government units at present including 6 metropolitan

**Table 1. Key characteristics of the study sites.**

| Characteristics | Bharatpur metropolitan city | Bidur municipality | Likhu rural municipality |
|---|---|---|---|
| District | Chitwan | Nuwakot | Nuwakot |
| Total population | 280502 | 54351 | 16852 |
| No. of households | 69035 | 12505 | 3629 |
| Population density (people/km$^2$) | 665.10 | 418.08 | 351.96 |
| No. of wards | 29 | 13 | 6 |
| No. of education institution | 220 | 64 | 22 |
| Literacy rate (%) | 98.5% | 65.1% | 58.8% |

cities, 11 sub-metropolitan cities, 276 municipalities, and 460 rural municipalities. This categorization of local government units is based on the minimum ceiling of the population of the local units, geography, ethnic/lingual/cultural density, bases of geographic accessibility, natural resources, institutional infrastructure, and potential of income/expenditure [18].

All the school-going students from the selected schools studying in grades 9 and 10 were included in the study. Those students who were not present during the day of data collection were excluded from the study.

## Study sample

The convenience sampling technique was used for study site selection and consecutive sampling was used for the selection of participants. All the students studying in grades 9 and 10 in selected schools were taken. We visited the respective schools and asked permission from the school administration to conduct the study. We informed the respondents and their parents about our study objectives and took their consent for the study.

## Study tools

A structured questionnaire was developed based on previous publications [12, 14] including the WHO STEPS survey questionnaire. The WHO STEPwise Approach to NCD Risk Factor Surveillance (STEPS) is a simple, standardized system for collecting, analyzing, and disseminating data on major NCD risk factors. The first section of the questionnaire included questions on participants' demographic characteristics (e.g., age, gender, grade of study, place of residence, family occupation, parental education, source of health information, and family history of diabetes or hypertension). The main questionnaire consisted of three sections on KAP. Each section included questions about Diabetes and hypertension.

We translated the questionnaire from English to Nepali language and back-translated it to English for its validation. The pretesting was done among 10% of the study sample, 40 students from grades 9 and 10 of Valmiki Shiksha Sadan, Bharatpur, Nepal. We modified the questionnaire based on pretesting results to make it more compatible with the respondents. We also checked the questionnaire for internal consistency (Cronbach's alpha = 0.7).

## Data analysis

The data collected through questionnaires were entered and analyzed by using SPSS (Statistical Package for Social Sciences) version 22. Knowledge questions were scored as 1 and 0 for correct and incorrect responses respectively. For example, for the multiple-choice question "Is Diabetes a communicable disease?", the response "No" scored 1 while the response "Yes" scored 0. We pooled "Do not know" responses with wrong answers and scored them as 0 which is a conventional practice as these responses come from the least knowledgeable

respondents or those respondents who really do not know [19]. Furthermore, if the respondents considered decreased physical activity, mental stress, family history, high salt intake, and obesity as the causes of hypertension, he/she was given 1 score for each of these causes and a score of 0 if they did not respond to them as the cause of hypertension. We scored attitude questions 1 for the positive attitude and 0 for the negative attitude or the neutral response.

The practice section of the questionnaire included questions on salt intake, servings of fruits and vegetables, and the amount of time spent on physical activities. We presented the findings from these questions as frequency percentage and mean values. The cut-off values for good practice in diet and physical activity were taken from WHO nutrition and physical activity guidelines. For example, WHO has recommended 5 or more servings of fruits and/or vegetables per day and at least 75 minutes of vigorous-intensity physical activity per week as a good behavioral practice to prevent NCDs [7].

We calculated relevant knowledge and attitude scores and categorized them into five categories (quintile groups) based on the percentage of maximum possible scores: "very poor" (0–20%), "poor" (21–40%), "satisfactory" (41–60%), "good" (61–80%) and "very good" (81–100%). The maximum possible scores for knowledge and attitude were 27 and 10 respectively.

Chi-square test was applied to compare the categorical variables among three local government units. One-way analysis of variance (ANOVA) was used to compare the mean knowledge and attitude scores among three different study areas. Logistic Regression analysis was carried out to identify the determinants of knowledge and attitude level of diabetes and hypertension. Independent variables included in the model were place of residence (local government unit), sex, grade of study, occupation, parental education, and family history of diabetes or hypertension. A dependent variable introduced to the model was knowledge and attitude level i.e., good (knowledge or attitude score >60% of total possible score) vs not good (knowledge or attitude score less than or equal to 60% of total possible score). Determinants having a screening significance of $p < 0.05$ in univariate analysis were selected for multivariate analysis. $P < 0.05$ was considered significant.

## Ethical consideration

Ethical approval for this study was taken from the Institutional Review Committee of Chitwan Medical College. All the participants were informed about the study and its objectives and informed consent was taken from the respondents' parents. Students at the selected schools were distributed a consent form to be signed by their parents and returned prior to data collection.

## Results

### Socio-demographic characteristics

Table 2 shows the socio-demographic characteristics of the respondents. Out of 380 respondents (school students enrolled in classes 9 and 10), 35.5% were residents of metropolitan city, 37.4% were from the urban municipality and 27.1% were from rural municipality. The mean age of respondents was 15.61± 0.99 years. Of the total respondents, 48.9% were female and 51.1% were male. Agriculture was the family occupation in the majority (42.1%) of participants. About 21.1% of the respondents had a family history of diabetes while 37.9% had a family history of hypertension. Among the socio-demographic variables, only mean age ($p < 0.05$), family occupation ($p < 0.001$), parental education level ($p < 0.001$), family history of diabetes ($p < 0.05$), and family history of hypertension ($p < 0.001$) were significantly different among the metropolitan city, urban municipality and rural municipality (Table 2).

**Table 2. Socio-demographic characteristics of the study population.**

| Variables | Metropolitan City (n = 135) | Urban Municipality (n = 142) | Rural Municipality (n = 103) | Total (n = 380) | p-value |
|---|---|---|---|---|---|
| Age (years) mean ± SD | 15.33± 0.88 | 15.77±0.92 | 15.77±1.14 | 15.61±0.99 | 0.001 |
| **Sex n (%)** | | | | | |
| Female | 64 (47.4) | 62 (43.7) | 60 (58.3) | 186 (48.9) | 0.079 |
| Male | 71 (52.6) | 80 (56.3) | 43 (41.7) | 194 (51.1) | |
| **Grade of study n (%)** | | | | | |
| Nine | 72 (53.3) | 67 (47.2) | 45 (43.7) | 184 (48.4) | 0.314 |
| Ten | 63 (46.7) | 75 (52.8) | 58 (56.3) | 198 (51.6) | |
| **Family occupation n (%)** | | | | | |
| Agriculture | 5 (3.7) | 59 (41.5) | 96 (93.2) | 160 (42.1) | |
| Government Employee | 41 (30.4) | 23 (16.2) | 3 (2.9) | 67 (17.6) | <0.001 |
| Private Employee | 14 (10.4) | 13 (9.2) | 2 (1.9) | 29 (7.5) | |
| Private Business | 75 (55.6) | 47 (33.1) | 2 (1.9) | 124 (32.5) | |
| **Parental education level n (%)** | | | | | |
| Illiterate | 0 (0.0) | 13 (9.2) | 21 (20.4) | 34 (8.9) | |
| Primary | 11 (8.1) | 43 (30.3) | 66 (64.1) | 120 (31.6) | <0.001 |
| Secondary | 76 (56.3) | 73 (51.4) | 15 (14.6) | 164 (43.2) | |
| Higher Secondary | 48 (35.6) | 13 (9.2) | 1 (1.0) | 62 (16.3) | |
| **Family history of diabetes n (%)** | | | | | |
| Yes | 35 (25.9) | 34 (23.9) | 11 (10.7) | 80 (21.1) | 0.01 |
| No | 100 (74.1) | 108 (76.1) | 92 (89.3) | 300 (78.9) | |
| **Family history of hypertension n (%)** | | | | | |
| Yes | 68 (50.4) | 53 (37.3) | 23 (22.3) | 144 (37.9) | <0.001 |
| No | 67 (49.6) | 89 (62.7) | 80 (77.7) | 236 (62.1) | |

## Sources of information on diabetes and hypertension

Fig 1 shows the sources of health information on diabetes and hypertension for the respondent students. The majority (77%) of the students from the metropolitan city responded to the Internet as the source of information on diabetes and hypertension followed by the school (68.9%). However, in urban municipality and rural municipality, the school was the source of health information among the majority of students (72.5% and 66% respectively) followed by the internet (43.7% and 26.2% respectively).

## Knowledge and attitude of students on diabetes and hypertension

Table 3 shows the mean knowledge score of respondents on diabetes and hypertension by socio-demographic characteristics. Among the respondents, those from the metropolitan city had significantly higher mean knowledge scores (13.35) than those from the urban municipality (9.36) and rural municipality (9.60). (p<0.001) Students with a family history of diabetes or hypertension scored significantly higher mean knowledge scores than those without such family history (p<0.001) (Table 3).

There was no statistically significant difference in the mean attitude scores of respondents belonging to different types of municipalities. Also, socio-demographic characteristics like sex, family occupation, parental education level, and family history of diabetes or hypertension did not result in a significant difference in the mean attitude scores of respondents. However, students enrolled in grade 10 scored significantly higher mean attitude scores than those in grade 9 (p<0.001) (Tables 4 and 5).

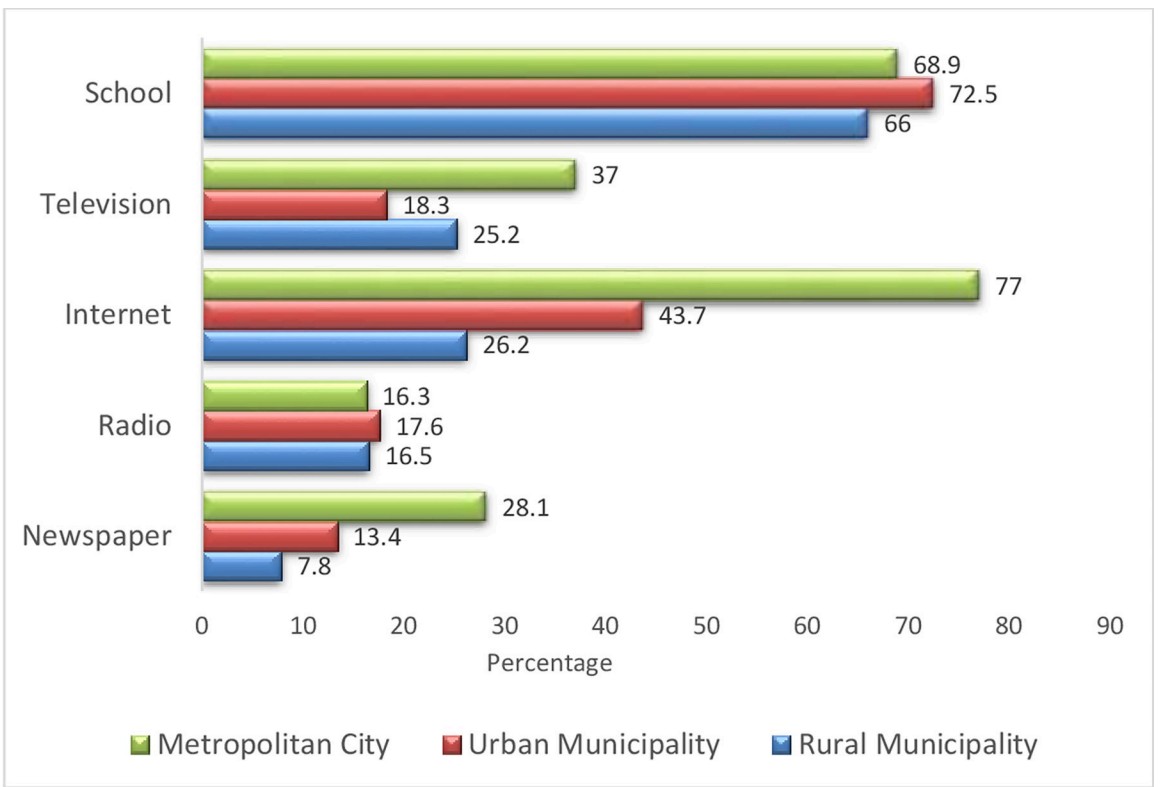

**Fig 1. Sources of information on diabetes and hypertension.**

Knowledge and attitude scores of respondents were classified into five categories (quintile groups) based on the percentage of maximum possible scores: "very poor" (0%-20%), "poor" (21%-40%), "satisfactory" (41%-60%), "good" (61%-80%) and "very good" (81%-100%). Fig 2 shows the level of knowledge of respondents on diabetes and hypertension. Out of the total respondents, the majority (48.4%) had a poor knowledge level. About two-thirds (68.1%) of respondents from the metropolitan city had satisfactory knowledge, while the majority from the urban municipality (66.9%) and rural municipality (63.1%) had poor knowledge of diabetes and hypertension.

However, the majority of the participants had a very good attitude towards diabetes and hypertension irrespective of the type of municipality they belong to (Figs 2 and 3).

### Practice behaviors on diabetes and hypertension

Table 6 shows the practice behaviors of respondents regarding diabetes and hypertension. Among the respondents, 9.7% responded that they always add salt to their food before eating or while eating, 71.1% responded that they add salt sometimes while 5.3% never add salt. Out of the total participants, 65.8% thought that they consume just the right amount of salt. 72.6% of participants from metropolitan city responded to consuming just the right amount of salt, while those consuming the right amount in urban municipality and rural municipality were 59.9% and 65.0% respectively.

About 29.1% of respondents from rural municipality consumed 5 and more servings of vegetables and/or fruits per day which was significantly higher in comparison to the percentage of participants from metropolitan city and urban municipality (11.1% and 23.3% respectively)

**Table 3. Respondents' knowledge score on diabetes and hypertension by socio-demographic characteristics.**

| Variables | Mean knowledge score | SD | p-value |
|---|---|---|---|
| **Municipality** | | | |
| Metropolitan city | 13.35 | 3.11 | <0.001 |
| Urban Municipality | 9.36 | 2.72 | |
| Rural Municipality | 9.60 | 3.08 | |
| **Sex** | | | |
| Female | 10.77 | 3.15 | 0.669 |
| Male | 10.92 | 3.79 | |
| **Grade of study** | | | |
| Nine | 10.61 | 3.37 | 0.218 |
| Ten | 11.06 | 3.60 | |
| **Family occupation** | | | |
| Agriculture | 9.64 | 3.04 | |
| Government Employee | 11.63 | 3.09 | <0.001 |
| Private Employee | 11.38 | 3.47 | |
| Private Business | 11.85 | 3.80 | |
| **Parental education level** | | | |
| Illiterate | 8.68 | 2.11 | |
| Primary | 9.93 | 3.44 | <0.001 |
| Secondary | 11.16 | 3.39 | |
| Higher Secondary | 12.95 | 3.26 | |
| **Family history of diabetes or hypertension** | | | |
| Yes | 11.50 | 3.74 | <0.001 |
| No | 10.23 | 3.14 | |

(p<0.05). There was a significant difference in average servings of fruits and/or vegetables per day among the participants from different places of residence (p<0.05).

## Effect of socio-demographic factors on knowledge and attitude levels of diabetes and hypertension

In univariate logistic regression analysis, place of residence: metropolitan city, occupation: other than agriculture, parental education: secondary and above and family history of diabetes or hypertension were significantly associated with good knowledge level. There were decreased odds of good attitude level among students studying in lower grade of study (grade nine) compared to a higher grade of study (grade ten). (OR: 0.35; 95% CI: 0.20–0.63) Variables having a p-value less than 0.05 were entered in the multivariable logistic regression analysis.

In multivariate logistic regression analysis, after adjusting for confounding factors, only the grade of study was the significant independent predictor of a student's good level of attitude toward diabetes and hypertension. Students who studied in grade nine (aOR: 0.32; 95% CI: 0.17–0.60, p<0.01) were found to have lower odds of good attitude compared to grade ten students (Table 7).

## Discussions

To the best of our knowledge, the present study is the first one in the Nepalese population that looked into urban-rural differences in knowledge, attitude, and practice of school-level students regarding diabetes and hypertension. In our study, family history of diabetes and hypertension was found more among the students of the metropolitan city than those from the

**Table 4. Attitude of respondents on diabetes and hypertension.**

| | Metropolitan City n (%) | Urban Municipality n (%) | Rural Municipality n (%) | Total n (%) | Chi-square p-value |
|---|---|---|---|---|---|
| Diabetes can be prevented with dietary modification. | | | | | 0.314 |
| Agree | 116 (85.9) | 111 (78.2) | 89 (86.4) | 316 (83.1) | |
| Disagree | 7 (5.2) | 15 (10.5) | 7 (6.8) | 29 (7.6) | |
| Neutral | 12 (8.9) | 16 (11.3) | 7 (6.8) | 35 (9.3) | |
| Regular exercise prevents Diabetes. | | | | | 0.399 |
| Agree | 110 (81.5) | 103 (72.5) | 75 (72.8) | 288 (75.8) | |
| Disagree | 14 (10.4) | 21 (14.7) | 17 (16.5) | 52 (13.8) | |
| Neutral | 11 (8.1) | 18 (12.8) | 11 (10.7) | 40 (10.4) | |
| If your family members or blood-related relatives have diabetes, you are also at risk of diabetes. | | | | | **0.001** |
| Agree | 65 (48.1) | 45 (31.7) | 32 (31.1) | 142 (37.3) | |
| Disagree | 47 (34.8) | 78 (54.9) | 63 (61.1) | 188 (49.5) | |
| Neutral | 23 (17.1) | 19 (13.4) | 8 (7.8) | 50 (13.2) | |
| Smoking worsens the complications of diabetes. | | | | | **<0.001** |
| Agree | 73 (54.1) | 92 (64.8) | 83 (80.6) | 248 (65.2) | |
| Disagree | 40 (29.6) | 31 (21.8) | 17 (16.5) | 88 (23.1) | |
| Neutral | 22 (16.3) | 19 (13.4) | 3 (2.9) | 44 (11.7) | |
| Regular blood sugar monitoring helps to control Diabetes. | | | | | 0.121 |
| Agree | 96 (71.1) | 110 (77.5) | 88 (85.4) | 294 (77.4) | |
| Disagree | 29 (21.5) | 22 (15.5) | 10 (9.7) | 61 (16.1) | |
| Neutral | 10 (7.4) | 10 (7.0) | 5 (4.9) | 25 (6.5) | |
| Diabetes can be prevented with dietary modification. | | | | | 0.314 |
| Agree | 116 (85.9) | 111 (78.2) | 89 (86.4) | 316 (83.1) | |
| Disagree | 7 (5.2) | 15 (10.5) | 7 (6.8) | 29 (7.6) | |
| Neutral | 12 (8.9) | 16 (11.3) | 7 (6.8) | 35 (9.3) | |
| Regular exercise prevents Diabetes. | | | | | 0.399 |
| Agree | 110 (81.5) | 103 (72.5) | 75 (72.8) | 288 (75.8) | |
| Disagree | 14 (10.4) | 21 (14.7) | 17 (16.5) | 52 (13.8) | |
| Neutral | 11 (8.1) | 18 (12.8) | 11 (10.7) | 40 (10.4) | |
| If your family members or blood-related relatives have diabetes, you are also at risk of diabetes. | | | | | **0.001** |
| Agree | 65 (48.1) | 45 (31.7) | 32 (31.1) | 142 (37.3) | |
| Disagree | 47 (34.8) | 78 (54.9) | 63 (61.1) | 188 (49.5) | |
| Neutral | 23 (17.1) | 19 (13.4) | 8 (7.8) | 50 (13.2) | |
| Smoking worsens the complications of diabetes. | | | | | **<0.001** |
| Agree | 73 (54.1) | 92 (64.8) | 83 (80.6) | 248 (65.2) | |
| Disagree | 40 (29.6) | 31 (21.8) | 17 (16.5) | 88 (23.1) | |
| Neutral | 22 (16.3) | 19 (13.4) | 3 (2.9) | 44 (11.7) | |
| Regular blood sugar monitoring helps to control Diabetes. | | | | | 0.121 |
| Agree | 96 (71.1) | 110 (77.5) | 88 (85.4) | 294 (77.4) | |
| Disagree | 29 (21.5) | 22 (15.5) | 10 (9.7) | 61 (16.1) | |
| Neutral | 10 (7.4) | 10 (7.0) | 5 (4.9) | 25 (6.5) | |

urban municipality and rural municipality. This finding is consistent with a study from India which reported the highest prevalence rate of self-reported diabetes in urban areas, the intermediate prevalence in peri-urban/slums, and the lowest in rural areas. In the same study, the urban residence was also reported to be one of the major risk factors for diabetes [20]. A recent study from South Africa also reported a higher prevalence of NCDs among urban areas in comparison to rural areas [21]. Urbanization has been an important factor contributing to the

**Table 5. Respondents' attitude scores on diabetes and hypertension by socio-demographic characteristics.**

| Variables | Mean attitude score | SD | p-value |
|---|---|---|---|
| **Municipality** | | | |
| Metropolitan city | 7.45 | 1.65 | 0.1 |
| Urban Municipality | 7.06 | 1.88 | |
| Rural Municipality | 7.47 | 2.05 | |
| **Sex** | | | |
| Female | 7.51 | 1.76 | 0.027 |
| Male | 7.09 | 1.92 | |
| **Grade of study** | | | |
| Nine | 6.85 | 1.97 | <0.001 |
| Ten | 7.72 | 1.64 | |
| **Family occupation** | | | |
| Farmer | 7.49 | 1.94 | |
| Government Employee | 6.85 | 1.84 | 0.095 |
| Private Employee | 7.03 | 1.94 | |
| Private Business | 7.35 | 1.72 | |
| **Parental education level** | | | |
| Illiterate | 6.85 | 2.42 | |
| Primary | 7.28 | 1.88 | 0.399 |
| Secondary | 7.31 | 1.84 | |
| Higher Secondary | 7.53 | 1.49 | |
| **Family history of diabetes or hypertension** | | | |
| Yes | 7.40 | 1.73 | 0.310 |
| No | 7.20 | 1.97 | |

growing burden of NCDs in low-and middle-income countries (LMICs) [22]. The higher prevalence of NCDs in urban areas has been attributed to various factors like low physical activity, consumption of inorganic food, exposure to environmental pollutants, and urban stressors compared to the rural population [21, 22].

In a previous study, rural men reported five times more physical activity as compared with urban men and rural women reported seven times more physical activity compared to urban women [23]. This finding is consistent with the present study as the amount of time spent on vigorous physical activity in a week is more among the respondents of the rural municipality compared to metropolitan city. However, contrary to the present study, the consumption of fruit and vegetables was considerably poor among the rural population compared to the urban population in the North Indian study [23]. In our study, respondents from rural communities had significantly high average servings of fruit and vegetables per day compared to the respondents from the metropolitan city and urban municipality. (p = 0.04) The families of most of the students from the rural municipality were involved in agricultural production as their major source of income, which might be the reason for the higher consumption of fruit and vegetables in the rural municipality.

Overall, the present study suggests that there is a relatively good attitude regarding diabetes and hypertension among students of Nepal despite the low level of knowledge. On the other hand, the mean knowledge scores were statistically significantly higher among students of the metropolitan city compared to urban and rural municipalities. (p<0.01) This finding is consistent with KAP studies on NCDs conducted among the general population in Malaysia and Pakistan [15, 24]. A possible reason for this low level of knowledge may be that there had

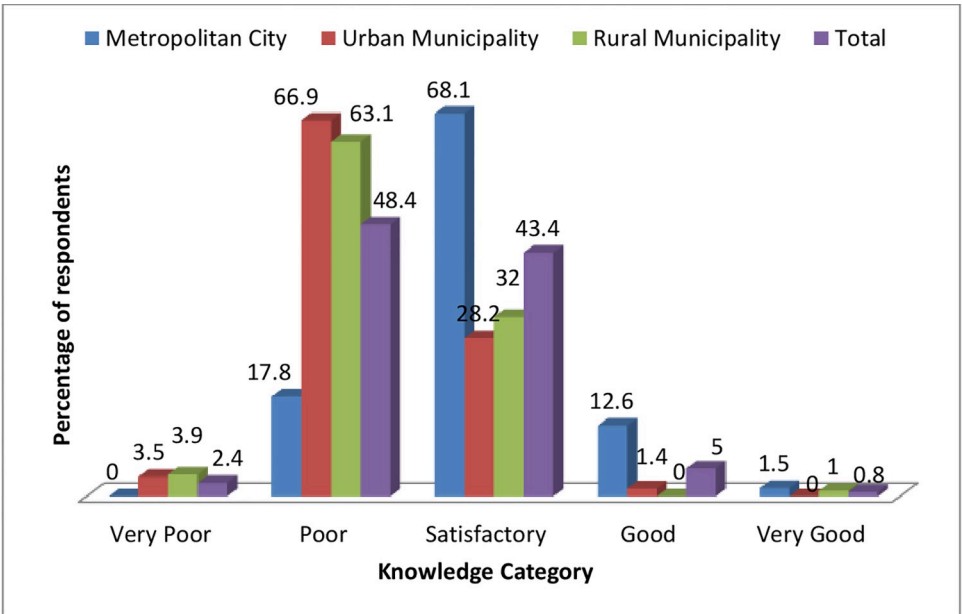

**Fig 2. Level of student's knowledge score on diabetes and hypertension.**

hardly been any regular school-based awareness campaign or health intervention program focusing on NCDs like diabetes and hypertension. Regarding attitude toward diabetes and hypertension, the majority of respondents had good attitude scores despite the low level of knowledge. The good attitude scores show that the majority of the students have perceived the risk of diabetes and hypertension and are supportive towards the prevention of NCDs. This finding is similar to the results of a study conducted among students of the University of Karbala in Iraq where more than 60% of students had very good attitude scores while only 32% of them had good knowledge scores. In our context, this result might be partially influenced by

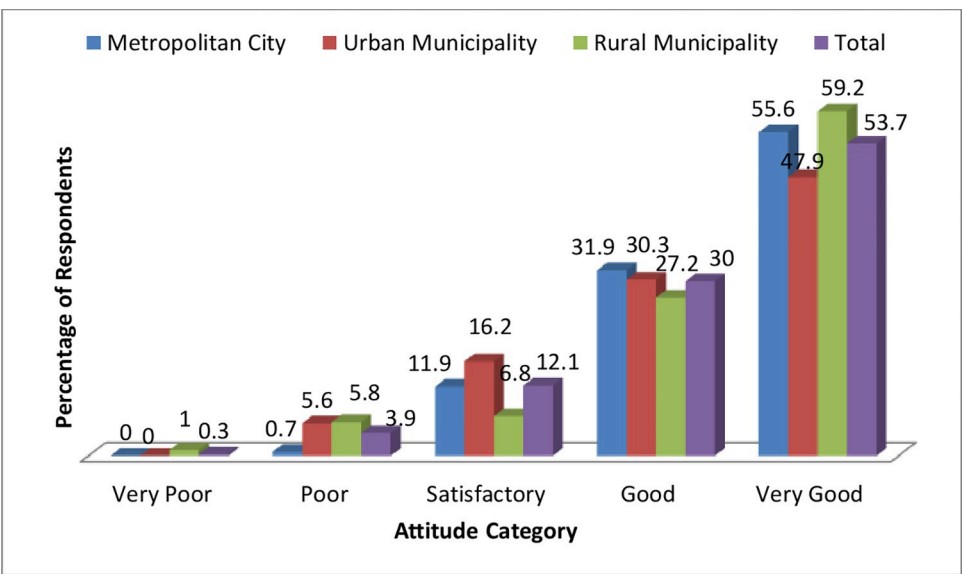

**Fig 3. Level of student's attitude score on diabetes and hypertension.**

**Table 6. Respondents' practice behaviors regarding diabetes and hypertension.**

| | Metropolitan City n (%) | Urban Municipality n (%) | Rural Municipality n (%) | Total n (%) | p-value |
|---|---|---|---|---|---|
| How often do you add salt to your food right before you eat it or as you are eating it? | | | | | 0.403 |
| Always | 9 (6.7%) | 19 (13.4%) | 9 (8.7%) | 37 (9.7%) | |
| Sometimes | 98 (72.6%) | 96 (67.6%) | 76 (73.8%) | 270 (71.1%) | |
| Never | 10 (7.4%) | 5 (3.5%) | 5 (4.9%) | 20 (5.3%) | |
| Don't Know | 18 (13.3%) | 22 (15.5%) | 13 (12.6%) | 53 (13.9%) | |
| How much salt do you think you consume? | | | | | 0.181 |
| Too much | 12 (8.9%) | 29 (20.4%) | 19 (18.4%) | 60 (15.8%) | |
| Just the right amount | 98 (72.6%) | 85 (59.9%) | 67 (65.0%) | 250 (65.8%) | |
| Too little | 21 (15.6%) | 25 (17.6%) | 15 (14.6%) | 61 (16.1%) | |
| Don't know | 4 (3.0%) | 3 (2.1%) | 2 (1.9%) | 9 (2.4%) | |
| How many servings of fruits and/or vegetables are served per day? | | | | | **0.009** |
| 1–2 | 41 (30.4) | 34 (23.9) | 28 (27.2) | 103 (27.1) | |
| 3–4 | 79 (58.5) | 75 (52.8) | 45 (43.9) | 199 (52.4) | |
| 5 and more | 15 (11.1) | 33 (23.3) | 30 (29.1) | 78 (20.5) | |
| Average servings per day | 3.4 ± 1.7 | 3.8 ± 1.9 | 4.0 ± 2.2 | 3.7 ± 1.9 | 0.049 |
| Amount of time spent on vigorous physical activity in a week (in minutes) | | | | | 0.87 |
| >75 minutes per week | 52 (38.5) | 58 (40.8) | 43 (41.7) | 153 (40.2) | |
| <75 minutes per week | 83 (61.5) | 84 (59.2) | 60 (58.3) | 227 (59.8) | |
| Average time spent per week (minutes) | 101.1 | 140.4 | 134.4 | 124.8 | 0.13 |

**Table 7. Multivariate logistic regression analysis showing predictors of knowledge and attitude levels (good vs. not good).**

| Dependent Variable | Independent Variable | COR (95% CI) | p-value | aOR (95% CI) | p-value |
|---|---|---|---|---|---|
| Knowledge | **Municipality** | | <0.01* | | 0.071 |
| | Metropolitan City | 1 | | 1 | |
| | Urban Municipality | 0.09 (0.02–0.38) | | 0.14 (0.02–0.77) | |
| | Rural Municipality | 0.06 | | 0.47 (0.02–12.5) | |
| | **Sex** | | 0.812 | | 0.166 |
| | Male | 1 | | 1 | |
| | Female | 0.37 (0.14–0.95) | | 0.48 (0.17–1.36) | |
| | **Occupation** | | <0.01* | | 0.116 |
| | Agriculture | 1 | | 1 | |
| | Non-Agriculture | 16.78 | | 6.73 (0.63–72.4) | |
| | **Education of parent** | | <0.01* | | 0.952 |
| | Primary and below | 1 | | 1 | |
| | Secondary and above | 4.62 (1.34–15.89) | | 0.96 (0.22–4.08) | |
| | **Grade of study** | | 0.22 | | 0.473 |
| | Ten | 1 | | 1 | |
| | Nine | 0.48 (0.19–1.19) | | 0.65 (0.20–2.13) | |
| | **Family history of diabetes or hypertension** | | 0.02* | | 0.053 |
| | Yes | 1 | | 1 | |
| | No | 0.32 (0.13–0.84) | | 0.33 (0.11–1.01) | |
| Attitude | **Grade of study** | | 0.03* | | < .001 |
| | Ten | 1 | | 1 | |
| | Nine | 0.35(0.20–0.63) | | 0.32 (0.17–0.60) | |

the tendency of students to please the interviewer/invigilator by agreeing to the statements of the questionnaire.

Among all the socio-demographic variables, place of residence (metropolitan city), occupation (other than agriculture), family history of diabetes or hypertension, and higher level of parental education were significantly associated with good knowledge levels on diabetes and hypertension. In a previous study conducted among university students in Saudi Arabia, female gender and family history of diabetes were associated with good knowledge scores on diabetes [25]. Another study reported that the global knowledge of adolescents on hypertension was significantly associated with place of residence (urban areas), higher grade of study, and family history of hypertension [26].

In line with the Multisectoral Action Plan for prevention and control of NCDs (2014–2020), the Government of Nepal adopted and implemented the WHO Package of Essential Non-communicable diseases interventions (PEN) in 2016 for the screening, diagnosis, treatment, and referral of cardiovascular diseases, diabetes, chronic respiratory diseases (asthma and COPD) and cancer. This package has four protocols: Prevention of heart attack, stroke, and kidney disease through integrated management of diabetes and hypertension; Health education and counseling on healthy behavior; Management of asthma and COPD; and assessment and referral of women with suspected cancer (breast, cervical) [27, 28]. This package was endorsed to enhance the coverage of relevant services to people with NCDs in primary care settings.

As the level of knowledge regarding diabetes and hypertension is not good among students of both urban and rural schools, school-based educational intervention programs will be very effective to improve the knowledge of students. A previously reported study from India has shown a significant impact of school-based educational programs which resulted in improvement of knowledge and behavioral changes related to diabetes among school students [29]. Initiatives like curriculum modification, interactive classroom sessions led by trained teachers, peer-led health activism, and lifestyle modification programs at the school level are necessary to improve knowledge and establish habits of healthy lifestyle among school students [29, 30].

## Limitations

The convenience sampling technique was used to select the study sites and the schools were not randomly selected for participation in the study. However, they were selected purposively to represent communities with different levels of urbanization. The inclusion of only four schools in the study limits the generalizability of the results. Some of the results were compared with studies conducted among adults of urban-rural communities due to a limited number of studies reporting rural-urban differences on KAP of school level students.

## Conclusions and recommendations

In comparison to students of the rural municipality, students of the urban municipality and metropolitan city had good knowledge about diabetes and hypertension. In general, there was a good attitude despite the low level of knowledge about diabetes and hypertension. This study shows an urgent need for school-based awareness programs focusing on non-communicable diseases and healthy lifestyle, especially in rural communities. More comprehensive inclusion of non-communicable diseases and their preventive measures in the school curriculum will help students to obtain good dietary habits and a healthy lifestyle from an early age. In addition to curriculum modification, extra-curricular activities like student-led health activism, regular school-based NCDs awareness campaigns, educational programs on diet and physical

activities, and student-volunteering in NCDs-related community activities can be effective to develop good knowledge as well as healthy habits among students.

## Supporting information

**S1 File.**
(PDF)

**S2 File.**
(PDF)

**S3 File.**
(SAV)

## Author Contributions

**Conceptualization:** Deekshanta Sitaula, Niki Shrestha, Santosh Timalsina.

**Data curation:** Deekshanta Sitaula, Niki Shrestha, Santosh Timalsina, Bandana Pokharel, Sachin Sapkota, Suchita Acharya, Rohit Thapa, Aarati Dhakal, Sarita Dhakal.

**Formal analysis:** Deekshanta Sitaula, Niki Shrestha, Santosh Timalsina.

**Methodology:** Deekshanta Sitaula, Niki Shrestha, Santosh Timalsina.

**Project administration:** Deekshanta Sitaula, Niki Shrestha, Santosh Timalsina, Bandana Pokharel, Sachin Sapkota, Suchita Acharya, Rohit Thapa, Aarati Dhakal, Sarita Dhakal.

**Resources:** Santosh Timalsina, Bandana Pokharel, Sachin Sapkota, Suchita Acharya, Rohit Thapa, Aarati Dhakal, Sarita Dhakal.

**Software:** Santosh Timalsina.

**Supervision:** Deekshanta Sitaula, Niki Shrestha, Santosh Timalsina.

**Validation:** Deekshanta Sitaula, Niki Shrestha, Bandana Pokharel, Sachin Sapkota, Suchita Acharya, Rohit Thapa, Aarati Dhakal, Sarita Dhakal.

**Visualization:** Deekshanta Sitaula, Niki Shrestha, Bandana Pokharel, Sachin Sapkota, Suchita Acharya, Rohit Thapa, Aarati Dhakal, Sarita Dhakal.

**Writing – original draft:** Deekshanta Sitaula.

**Writing – review & editing:** Niki Shrestha, Santosh Timalsina.

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
