## [Decision Letter · Decision Letter 0]

27 Apr 2022

PONE-D-22-03214Knowledge, attitude and practice regarding selected non-communicable diseases (NCDs) among school students of Nepal: A rural vs. urban studyPLOS ONE

Dear Dr. Sitaula,

Thank you for submitting your manuscript to PLOS ONE. After careful consideration, we feel that it has merit but does not fully meet PLOS ONE’s publication criteria as it currently stands. Therefore, we invite you to submit a revised version of the manuscript that addresses the points raised during the review process.

Dear authors, both the reviewers have submitted there reports and major revision is required to the manuscript. Incorporate the changes suggested by highlighting the addition and resubmit. 

We look forward to receiving your revised manuscript.

Kind regards,

Rohit Ravi, Ph.D.

Academic Editor

PLOS ONE

Journal Requirements:

Reviewers' comments:

Reviewer's Responses to Questions

**Comments to the Author**

1. Is the manuscript technically sound, and do the data support the conclusions?

Reviewer #1: Yes

Reviewer #2: Yes

2. Has the statistical analysis been performed appropriately and rigorously? 

Reviewer #1: Yes

Reviewer #2: Yes

3. Have the authors made all data underlying the findings in their manuscript fully available?

Reviewer #1: Yes

Reviewer #2: Yes

4. Is the manuscript presented in an intelligible fashion and written in standard English?

Reviewer #1: No

Reviewer #2: Yes

5. Review Comments to the Author

Reviewer #1: The authors of this study provided beneficial evidence on the knowledge, attitude, and practice (KAP) regarding two major non-communicable diseases (NCDs) among school students of Nepal. Since the NCDs is a major public health concern in this country, proper evaluation of the knowledge and education among the youth could leverage the health status in the future. Although the study is designed and drafted well, the manuscript could benefit some comments to improve its strengths and readability. My suggestions and comments are provided as follows.

1. General: a language and grammar edit are essential on this manuscript due to many errors through the text. Also, the citations are doubled and need revision.

2. Title: since this study evaluates KAP regarding only diabetes and hypertension, authors may replace the term “selected NCDs” with the two mentioned conditions to present a clear title of study.

3. Abstract: although this part summarizes the manuscript very well, the prepared abstract is too long and should be briefer for a faster review of the manuscript. Therefore, it is suggested that authors drop unnecessary details of the abstract.

4. Introduction: the provided statistics of the NCDs’ burden is outdated and authors may use more updated and recent estimations and studies to introduce the importance of NCDs. One of the robust studies in this regard is the Global Burden of Disease Study which its recent iteration known as GBD 2019 provides the most statistics (https://vizhub.healthdata.org/gbd-compare/).

5. Methods, Study Settings and Participants: the selected three type of study settings is misleading. It is highly suggested that authors select only two major population of urban versus rural students to understand the difference easier and also to benefit the health policy makers.

6. Methods, Study Tools: the referred previous studies need proper citations. Also, the STEPS survey should be elaborated for the audience who are not familiar with this study.

7. Methods, Data analysis: all recruited criteria from WHO methods in this study should be clearly explained.

8. Results: since this section presents a huge amount of results, it is highly suggested that authors make a revision on this section and rearrange them in 3-4 main subsections to make following the results easier for the audience. Also, the provided tables and figures are appropriate and authors may delete some of the text and refer the reader to the tables and summarize the section.

9. Discussion: it is suggested that authors begin this section with a general interpretation of the findings instead of providing numbers of the results section.

10. Discussion: one paragraph is needed in this section discussing the ongoing programs regarding the control of NCDs in Nepal and discussing its limitations and the gaps, since it is necessary when authors investigate the KAP regarding NCDs in this manuscript.

Reviewer #2: This school-based cross-sectional study by Deekshanta Sitaula, et al., 2022 assessed the knowledge, attitude, and practice of rural and urban school students regarding diabetes and hypertension in Nepal, and attempted to determine the differences in the knowledge, attitude and practice of students from rural vs. Urban communities.

The researchers used a pre-tested structured questionnaire developed based on previous publications including WHO STEPS survey questionnaire. Data were collected from participants between May 1, 2021 to June 30, 2021.

Dependent (outcome) variable: knowledge and attitude level

Independent variables: place of residence (local government unit), sex, grade of study,

occupation, parental education and family history of diabetes or hypertension

The data collected from 380 respondents were analyzed in the Statistical Packages for Social Sciences (SPSS) version 20.0. Logistic regression analysis was carried out to determine the determinants of knowledge and attitude regarding diabetes and hypertension.

Results revealed that respondents from the metropolitan city had significantly higher mean knowledge scores than those from urban and rural municipality (p<0.001) while there was no

significant difference in mean attitude scores. There was significantly higher daily consumption of fruits and vegetables among the participants from rural municipality compared to metropolitan city and urban municipality (p<0.01) while no significant difference was seen in salt consumption and time spent on physical activity. In univariate regression analysis, place of residence, family occupation, parental education and family history of diabetes and hypertension were significantly associated with good knowledge level. In multivariate analysis, only higher grade of study (grade 10 in comparison to grade 9) was an independent predictor of student's good attitude level.

The authors made all data underlying the findings fully available. The data was tested for representativeness, analyzed using descriptive and inferential statistics which were rigorous and appropriate.

Discussions of the results were robust, citing similar studies conducted both within and outside Nepal.

Conclusions are in line with the findings

Writing quality and clarity: Satisfactory

Other observations:

1. Limitations of the study: The authors did well to mention the limitations of the study but they fell short of suggesting how these limitations should be addressed by future studies going forward

2. Inclusion/exclusion criteria should be more detailed

References: The manuscript employed the use of Harvard style referencing but requires editing to correct some errors noticed e.g., Listing of references: Shouldn’t this be in alphabetical order? Shouldn’t the journal name be italics? Shouldn’t the list of authors that are more than 5 be reflected as et al?

I suggest the authors should revise Harvard referencing style and make necessary corrections.

6. PLOS authors have the option to publish the peer review history of their article (what does this mean?). If published, this will include your full peer review and any attached files.

Reviewer #1: **Yes: **Sina Azadnajafabad, MD, MPH

Reviewer #2: **Yes: **Haruna Ismaila ADAMU, MD; MPH; PhD; MACE

---

## [Author Response · Author response to Decision Letter 0]

11 May 2022

Answer to comments of reviewer 1: 

Thank you for the comments.

1. General: a language and grammar edit are essential on this manuscript due to many errors through the text. Also, the citations are doubled and need revision.

• Necessary grammatical edits have been done after revision. We have rechecked the citations and we didn't find any duplications. However, some authors are common in more than one reference as they have published series of articles on NCDs in Nepal.

2. Title: since this study evaluates KAP regarding only diabetes and hypertension, authors may replace the term “selected NCDs” with the two mentioned conditions to present a clear title of study.

• In the title, "selected NCDs" has been replaced with "diabetes and hypertension". 

3. Abstract: although this part summarizes the manuscript very well, the prepared abstract is too long and should be briefer for a faster review of the manuscript. Therefore, it is suggested that authors drop unnecessary details of the abstract.

• Some parts of the abstract which are less necessary have been dropped. 

4. Introduction: the provided statistics of the NCDs’ burden is outdated and authors may use more updated and recent estimations and studies to introduce the importance of NCDs. One of the robust studies in this regard is the Global Burden of Disease Study which its recent iteration known as GBD 2019 provides the most statistics.

• The data used were from latest WHO country profile 2018. Newest data on NCDs from GBD 2019 have been added in introduction section. 

5. Methods, Study Settings and Participants: the selected three type of study settings is misleading. It is highly suggested that authors select only two major population of urban versus rural students to understand the difference easier and also to benefit the health policy makers.

• We discussed with our statistician and supervisors regarding the adjustment in the type of study settings. The study settings in this study represents three levels of administrative divisions of Nepal, the major basis of which is level of urbanization. So, they have advised that these settings represent the status of KAP in different local levels of the country. These findings will be beneficial in policy making as well as health program designing, as local government has an important role in designing and conducting the health programs in local levels. 

6. Methods, Study Tools: the referred previous studies need proper citations. Also, the STEPS survey should be elaborated for the audience who are not familiar with this study.

• The referred previous studies have been cited.

• A description of STEPS survey has been added in this section.

7. Methods, Data analysis: all recruited criteria from WHO methods in this study should be clearly explained.

• Only few questions on physical activity, vegetable and fruit consumption and salt intake have been taken from STEPS survey to include in the practice section of the KAP questionnaire. They have been presented as mean values, frequency and percentage as per the WHO STEPS survey. This has been mentioned in the data analysis section.

8. Results: since this section presents a huge number of results, it is highly suggested that authors make a revision on this section and rearrange them in 3-4 main subsections to make following the results easier for the audience. Also, the provided tables and figures are appropriate and authors may delete some of the text and refer the reader to the tables and summarize the section.

• The three sub-sections "knowledge scores", "attitude scores" and "level of knowledge and attitude" have been merged into one- "knowledge and attitude of students on diabetes and hypertension".

• The results section has been rearranged in 5 main sub-sections- "Socio-demographic characteristics", "sources of information on diabetes and hypertension", "knowledge and attitude of students on diabetes and hypertension", "Practice behaviors on diabetes and hypertension" and "Effects of socio-demographic factors on knowledge and attitude on diabetes and hypertension. 

9. Discussion: it is suggested that authors begin this section with a general interpretation of the findings instead of providing numbers of the results section

• The numerical figures have been removed from the discussion section and only the general interpretation of the findings has been mentioned. 

10. Discussion: one paragraph is needed in this section discussing the ongoing programs regarding the control of NCDs in Nepal and discussing its limitations and the gaps, since it is necessary when authors investigate the KAP regarding NCDs in this manuscript.

• A paragraph explaining about ongoing programs regarding the control of NCDs in Nepal has been added in the discussion section. 

Answers to comments of reviewer 2:

Thank you for the feedback.

2. Inclusion/exclusion criteria should be more detailed.

• Consecutive sampling was used and all the students of class 9 and 10 of the selected schools were included in the study. Only the students who were absent at the time of data collection were excluded from the study. There were no other exclusion criteria. 

References:

References: The manuscript employed the use of Harvard style referencing but requires editing to correct some errors noticed e.g., Listing of references: Shouldn’t this be in alphabetical order? Shouldn’t the journal name be italics? Shouldn’t the list of authors that are more than 5 be reflected as et al?

I suggest the authors should revise Harvard referencing style and make necessary corrections.

• Vancouver style referencing has been used in the manuscript as per the journal submission guideline.

• The references have been listed numerically as per their citation in the manuscript.

• It has been stated that italics may not be required in Vancouver style.

• In Vancouver, the list of authors that are more than 6 are reflected as et al. This has been used in the manuscript.

Journal Requirements:

• PLOS ONE's style requirements have been checked and the manuscript has been adjusted accordingly.

• Mention of parental consent has been added in the method section.

---

## [Editor Report · Decision Letter 1]

7 Jun 2022

Knowledge, attitude and practice regarding diabetes and hypertension among school students of Nepal: A rural vs. urban study

PONE-D-22-03214R1

Dear Dr. Sitaula,

We’re pleased to inform you that your manuscript has been judged scientifically suitable for publication and will be formally accepted for publication once it meets all outstanding technical requirements.

Kind regards,

Rohit Ravi, Ph.D.

Academic Editor

PLOS ONE

---

## [Editor Report · Acceptance letter]

22 Aug 2022

PONE-D-22-03214R1 

Knowledge, attitude and practice regarding diabetes and hypertension among school students of Nepal: A rural vs. urban study 

Dear Dr. Sitaula:

I'm pleased to inform you that your manuscript has been deemed suitable for publication in PLOS ONE. Congratulations! Your manuscript is now with our production department. 

Kind regards, 

on behalf of

Dr. Rohit Ravi 

Academic Editor

PLOS ONE